# Graph-based machine learning improves just-in-time defect prediction

Jonathan Bryan[1]☯*, Pablo Moriano [2]☯*

**1** AT&T Cybersecurity, AT&T, Atlanta, GA, United States of America, **2** Computer Science and Mathematics Division, Oak Ridge National Laboratory, Oak Ridge, TN, United States of America

☯ These authors contributed equally to this work.
* jz699j@att.com (JB); moriano@ornl.gov (PM)

**Data Availability Statement:** All data are available from the GitHub repository (https://github.com/lining-nwpu/JiTReliability).

**Funding:** This manuscript has been authored by UT-Battelle, LLC under ContractNo. DE-AC05-00OR22725 with the U.S. Department of Energy.

## Abstract

The increasing complexity of today's software requires the contribution of thousands of developers. This complex collaboration structure makes developers more likely to introduce defect-prone changes that lead to software faults. Determining when these defect-prone changes are introduced has proven challenging, and using traditional machine learning (ML) methods to make these determinations seems to have reached a plateau. In this work, we build contribution graphs consisting of developers and source files to capture the nuanced complexity of changes required to build software. By leveraging these contribution graphs, our research shows the potential of using graph-based ML to improve Just-In-Time (JIT) defect prediction. We hypothesize that features extracted from the contribution graphs may be better predictors of defect-prone changes than intrinsic features derived from software characteristics. We corroborate our hypothesis using graph-based ML for classifying edges that represent defect-prone changes. This new framing of the JIT defect prediction problem leads to remarkably better results. We test our approach on 14 open-source projects and show that our best model can predict whether or not a code change will lead to a defect with an F1 score as high as 77.55% and a Matthews correlation coefficient (MCC) as high as 53.16%. This represents a 152% higher F1 score and a 3% higher MCC over the state-of-the-art JIT defect prediction. We describe limitations, open challenges, and how this method can be used for operational JIT defect prediction.

## 1 Introduction

Software quality assurance, including source code inspection and testing, has become increasingly necessary for building high-quality software [1]. Software defects, or bugs, are detrimental to software quality and have a negative economic and reputational impact on software stakeholders, especially when they lead to software failures [2]. Thus, there is a huge incentive to detect likely software defects as early as possible in the development process. Reducing the number of software defects through quick and automatic identification would lead to the production of better software by improving its usability and reducing costs associated with maintenance.

The publisher, by accepting the article for publication, acknowledges that the U.S. Government retains a non-exclusive, paid up, irrevocable, world-wide license to publish or reproduce the published form of the manuscript, or allow others to do so, for U.S. Government purposes. The DOE will provide public access to these results in accordance with the DOE Public Access Plan (http://energy.gov/downloads/doe-public-access-plan). This research was sponsored in part by Oak Ridge National Laboratory's (ORNL's) Laboratory Directed Research and Development program and by the DOE, Office of Science, Office of Workforce Development for Teachers and Scientists (WDTS) under the Scientific Undergraduate Laboratory Internship (SULI) program. Pablo Moriano acknowledges support from ORNL's Artificial Intelligence initiative. The funders had no role in study design, data collection and analysis, decision to publish, or preparation of the manuscript. There was no additional external funding received for this study.

**Competing interests:** The authors have declared that no competing interests exist.

Previous research on software quality assurance focuses on either module-level [3] or Just-In-Time (JIT) defect prediction [4]. The module-level approach uses machine learning (ML) models trained on historical data obtained from software characteristics, including code churn, change metadata, and complexity metrics [5]. Defect prediction models detect defect-prone software modules (e.g., files [6], subsystems [7]). Defect prediction models are then used to identify software modules that likely contain faulty code. These models can also help prioritize software quality assurance efforts, such as code reviews and pre-release testing. The JIT approach, in contrast, focuses on change-level defect prediction. This means that the focus is on software changes (i.e., commits) rather than on modules.

JIT has important advantages over module-level defect prediction [4]. First, it reduces defect detection time: JIT predictions are obtained when changes are ready to be committed, *before* the software has been deployed. Second, it provides attribution: JIT predictions are linked to the author of the change rather than a group of authors. Lastly, it produces finer-grained predictions: JIT predictions spotlight specific changes, which are often smaller than coarser-grained prediction modules. Therefore, predicting defect-prone changes using JIT is preferred over module-level defect prediction.

Current JIT defect prediction models use software characteristics to inform commonly used, supervised ML models. Traditional features used in this task are related to the diffusion, size, purpose, history, and experience dimensions of the changes [4]. Recent models also add context to these features by leveraging the semantic information and syntactic structure hidden in source code [8]. Once this set of features has been computed for the targeted software commits, different ML models are used for JIT defect prediction, including logistic regression [9] and more sophisticated models, such as ensembles [10] and deep learning [11–13]. More recently, new features based on representing code semantics using word embeddings to map change sequences into numeric vectors have been proposed [14].

Obtaining large amounts of accurate historical data is a prerequisite for good performance in JIT defect prediction [15]. However, this data can be difficult to obtain because the nature of code commits/changes tends to evolve during the development cycle, which can impact the performance of JIT defect prediction [16]. In addition, as shown recently [10], even when using sophisticated ML models, such as ensembles, the achievable performance for JIT defect prediction still has much room for improvement (i.e., it currently reaches about 31% average F1 score and 51% Matthews correlation coefficient (MCC) for predicting early exposed defects).

Here, we introduce a novel framework for JIT defect prediction using contribution graphs [17] and graph-based ML [18]. Contribution graphs are bipartite graphs in which nodes represent developers and modules (source code files in our case). Edges in the contribution graph capture interactions between developers and modules, thereby representing software changes. We label edges in these graphs to distinguish clean commits from bug-introducing commits using the Sliwerski-Zimmermann-Zeller (SZZ) algorithm [19]. We then extract features from the contribution graph using (1) centrality metrics (Setting 1) and (2) community assignments and node embeddings (Setting 2) [20]. These two feature sets are then used to inform ML algorithms and classify code changes.

Our approach is novel for JIT defect prediction in that it assigns a probability score to each new code change (i.e., an unlabeled edge in the graph) that indicates the likelihood of that change being defect-prone. We operationalize this idea using edge classification. Edge classification refers to the problem of classifying unknown edge labels in a graph [21]. Here, the notion of an edge appearing in the future is quantified as a score that measures the likelihood of it being a defect-prone change. We show the potential of using this approach with higher classification results (i.e., with a 152% higher F1 score and a 3% higher MCC) compared to a

recent benchmark on JIT defect prediction on early exposed defects over 14 large open-source projects [10].

The main contributions of this paper are as follows. First, we investigate the use of graph-based ML for JIT defect prediction. The core of our contribution is data modeling by using contribution graphs. In particular, we leverage contribution graphs as a modeling framework to capture the changes that developers make to software files. We used this abstraction as the basis of edge classification. Specifically, from the contribution graphs, we extract graph-related features that inform edge classification models when classifying defect-prone changes. Second, we perform an in-depth evaluation of these graph-based ML models using (1) centrality metrics (Setting 1) and (2) community assignments and node embedding features (Setting 2) while taking into account the unbalanced nature of the dataset. Lastly, we compare these graph-based models with traditional ML models, including 11 state-of-the art JIT defect detection methods. Our results show that the graph-based approach provides a 152% higher F1 score and a 3% higher MCC than the state-of-the-art. We are sharing the data [22] and code [23] used in this research so that our results can be reproduced.

## 2 Related work

Our research is informed by past work in network analysis for software engineering, JIT defect prediction, and graph-based ML. Here, we provide an overview of this related work.

### 2.1 Network analysis in software engineering

Network analysis is used to model the interactions of software elements, including between software dependencies (i.e., the dependency graph [24]) and between developer and software modules (i.e., the contribution graph [17]). This modeling framework has also been used to predict failures in files within a closed networking software project [25], examine the relationship between ownership measures and software failures [26], quantify the impact of network analysis metrics as indicators of software vulnerabilities [27], and estimate insider threat risk in a version control system (VCS) [28].

One significant difference between our work and other studies using network analysis is that our work uses features derived from the contribution graph for JIT defect prediction. In doing so, we frame the problem of introducing defect-prone changes as the likelihood that unseen edges introduce them. We explore two kinds of network properties: (1) topological properties (Setting 1) and (2) community assignations and node embeddings (Setting 2) in the contribution graph.

### 2.2 SZZ algorithm

The SZZ algorithm is the primary algorithm used to identify defect-prone changes (i.e., bug-introducing commits) in a software repo. Using the SZZ algorithm, practitioners can identify individual commits that introduce defect-prone changes. The SZZ algorithm was introduced by Śliwerski et al. [19] and was originally conceived for centralized VCSs, such as CVS and its corresponding commit practices. Later iterations of the SZZ algorithm made it operational for distributed VCSs, such as `git`.

The SZZ algorithm uses two sources of data: (1) bug reports (BRs) from an issue tracker system (ITS), such as Jira or BugZilla, and (2) historical change logs from a VCS. The SZZ algorithm has two main steps. In the first step, BRs are linked to defect-fixing changes (i.e., bug-fixing commits). This is achieved by finding explicit calls to BRs in commit messages by using regular expressions. In the case in which the ITS does not allow the identification of bug fixes, commit messages using the word *fix* (or similar) are used as a proxy for identifying defect-

fixing changes. In the second step, once defect-fixing changes are identified, the SZZ algorithm traces back the modified lines in the source code. Specifically, for each of the identified defect-fixing changes, the SZZ algorithm uses `git blame` to identify previous commits that made changes to those specific lines of code. Git blame also extracts the revision and the last author to modify those lines. This means that the output of git blame contains a set of candidate commits that may have introduced the defect. From this set of candidates, the SZZ algorithm determines whether or not any commits can be discarded as a defect-prone change. Borg et al. provide a detailed description of the heuristic used in the SZZ algorithm [29].

## 2.3 JIT defect prediction

JIT defect prediction consists of four main steps. First, JIT uses the SZZ algorithm [19] to label previous changes obtained from a VCS as defect prone or not. The SZZ algorithm is the primary algorithm used to identify defect-prone changes (i.e., bug-introducing commits) in a software repo. Second, it quantifies change metrics that characterize changes in the code. Third, it learns a ML classifier based on the previously computed labeled changes and their metrics. Finally, JIT defect prediction uses the learned classifier to predict if new code changes are defect-prone.

An important aspect of JIT defect prediction is identifying and extracting the independent variables used to build JIT defect prediction models. Generally, there are two approaches for doing that: feature engineering and feature learning. In the former, features are designed manually. In the latter, features are learned automatically via algorithms as feature representations from the data.

Early work on JIT defect prediction through feature engineering focused on using features derived from software change metrics as those directly computed from code changes. They include variables related to diffusion (e.g., number of modified subsystems), size (e.g., lines of code added), history (e.g., number of developers who modified the files), and experience (e.g., developer experience) [30]. The origin of JIT defect prediction through feature engineering is usually attributed to Mockus and Weiss [31]. Working with a large, closed telecommunication code base, Mockus and Weiss introduced the idea of quantifying software change properties to predict defects for initial maintenance requests (IMRs) when the IMRs consist of multiple changes. Kim et al. [32] focused on predicting individual defect-prone changes and applied this method to a variety of open-source code bases. Kamei et al. [4] extended previous work by applying it to open-source and commercial code bases across multiple industries. Jiang et al. [33] introduced a personalized defect prediction approach by building a separate prediction model for each developer. Kononenko et al. [34] added features extracted from code review databases, thereby leading to an increase in the explanatory power of JIT defect prediction models. Kamei et al. [35] evaluated the performance of JIT defect prediction in projects still in their initial development phases. In doing so, they showed that JIT models trained using data from projects with sufficient history are viable candidates for JIT defect prediction in projects with limited historical data.

Early work on JIT defect prediction via feature learning focused on estimating feature representations by extracting the amount of information in the commit message. Deep learning approaches, and deep neural networks in particular, are predominantly used to learn feature representations for JIT defect prediction. For example, Yang at al. [11] built a set of expressive features from a set of initial changes by leveraging a deep belief network algorithm. Later, they trained an ML classifier on selected features. Barnett et al. [36] mined commit message content by using SpamBayes classifiers [37] as filters. More recently, Hoang et al. [12] proposed an end-to-end deep learning framework, named DeepJIT, which automatically extracts features

from code changes and commit messages and uses them to classify defects. Xu et al. [38] proposed a cross-triplet deep feature embedding method, called CDFE, for cross-app JIT defect prediction in mobile apps. The CDFE method incorporates a cross-triplet loss function into a deep neural network to learn high-level feature representation for the cross-app data. This loss function shortens the distance of commits with similar labels while lengthening the distance between commits with different labels. Zhuang et al. [14] proposed a method to represent code semantics based on Abstract Syntax Tress (ASTs). This method works by comparing the AST of source code before and after a change for extracting change sequences that are then mapped to a numeric vector by using word-embedding models.

Table 1 summarizes closely related previous work on JIT defect prediction that uses feature learning. For a comprehensive survey on JIT defect prediction, we refer the reader to the work by Zhao et al. [30].

To the best of our knowledge, apart from features derived through feature engineering (based on software code change metrics) and features derived using feature learning, no prior studies have investigated the use of features derived from the contribution graphs to inform graph-based ML classifiers to predict the risk of introducing defect-prone changes.

## 2.4 Graph-based ML

Graph-based ML refers to the use of graph-based related features to train ML algorithms [39]. Graph-based features can be highly predictive, thereby adding value to existing ML models. Applications of graph-based ML span multiple industries, including attribute prediction in social networks [40], bot detection [18], understanding the dynamics of opioid doctor shopping [41], and cybersecurity applications, such as detection of lateral movement in enterprise computer networks [42].

Graph-based ML is based on extracting structural features from graphs. These features can be obtained from the structure of the graphs or by using representation learning (graph embeddings). The former refers to traditional structural properties, such as a node's degrees and/or centrality metrics [43]. The latter refers to encoding structural information of individual nodes into a low-dimensional vector space. Graph embedding methods are flexible because they can adapt during the learning process, as opposed to purely structured features that require feature engineering. Graph embedding methods are classified based on the algorithm used for the encoding [44]. This classification includes matrix factorization [45, 46], random

**Table 1. Summary of JIT defect prediction works by using feature learning.**

| Study | Year | Primary Topic | Constraint |
|---|---|---|---|
| Yang et al. [11] | 2015 | Proposed Deeper, which consists of a deep belief network and a logistic regression classifier for JIT defect prediction | Deeper does not fully exploit the true benefits of deep learning because it uses the same set of traditional features derived from feature engineering. |
| Barnett et al. [36] | 2016 | Investigated the benefits of adding commit message features for JIT defect prediction | Leveraging commit message proneness to be defective by using a SpamBayes classifier may produce overly specific scores for systems to which theywill be applied. |
| Hoang et al. [12] | 2019 | Introduced DeepJIT, an end-to-end JIT defect prediction model that learns feature representations from tokenized software changes andcommit messages | Because code is written using an open, rapidly changing vocabulary, word embeddings may introduce noise and then cause a negative impact on the performance of the cross-project defect prediction. |
| Xu et al. [38] | 2021 | Designed CDFE, a deep neural network with triplet loss function for cross-project JIT defect prediction in mobile apps | Using labeled commit data from other mobile apps to assist the labeling of the mobile app at hand may be impacted by the imbalanced nature of JIT defect prediction. |
| Zhuang et al. [14] | 2022 | Introduced ACE based on Abstract Syntax Trees (ASTs) by comparing the AST of source code before and after a change and computing change sequences that are mapped to word embeddings | Experiments conducted in projects coded on Java. AST nodes of different programming languages may differ, thereby influencing JIT defect prediction performance. |

walks [20, 47], or deep learning methods [48, 49]. The embedding choice usually depends on the application [50].

Our work is novel in that it compares second-order proximity metrics, which describe the proximity of node pairs and their neighborhood's structure using centrality metrics and a random walk graph embedding (i.e., `node2vec` [20]). `node2vec` performs biased random walks by trading the bias between breadth-first and depth-first search. `node2vec` is parameterized using walk length, context size, and bias weights, and its embeddings ensure that nodes with common neighbors tend to appear close in the embedding space.

## 3 Methods

This section describes the mathematical frameworks and data sources used to perform this research. The proposed method for JIT defect prediction leverages contribution graphs to learn classifiers that distinguish regular commits from defect-prone changes. We extracted graph-based features to quantify the risk that new changes will introduce defects. Performance is measured by the the ability of the algorithm to distinguish between regular commits and defect-prone changes, i.e., edge classification.

An overview of the proposed framework is presented in Fig 1. Our method is composed of three main phases: dataset generation, training, and testing. In the dataset generation phase, we build a labeled commit dataset by combining the extracted graph features from the contribution graph (see Section 3.1) and bug-inducing commits produced by the SZZ algorithm. During the training phase, we process the training data (see Section 3.3) by conducting data preparation (see Section 3.3.1), model training (see Section 3.3.2), and selection (see Section 3.3.3) on a subset of classifiers (see Section 3.4). During the testing phase, we prepare the data from given code changes and feed it into the trained model (see Section 3.3.4). Predictions from the classifiers are used to estimate the quality of the predictions (see Section 3.5). We associated each step in the proposed method (and their subsequent section numbers) with the corresponding phase in Fig 1.

### 3.1 Graph modeling

Each of the steps in our graph modeling framework are detailed below.

**3.1.1 Contribution graph.** We model contribution graphs as undirected bipartite graphs made of developers (top nodes) and source code files (bottom nodes). We let $\mathcal{H}_\top$ represent the set of top nodes, or developers; and we let $\mathcal{H}_\perp$ be the set of bottom nodes, or files. Note that $\mathcal{H}_\top$ and $\mathcal{H}_\perp$ are a disjoint set of nodes. Let $\mathcal{V} = \mathcal{H}_\top \cup \mathcal{H}_\perp$ be the set of contribution graph nodes. Let $\mathcal{W} = \{\omega_{ij} : (i,j) \subseteq \mathcal{H}_\top \times \mathcal{H}_\perp\}$ be the incident matrix of weights, $\omega_{ij}$, that captures

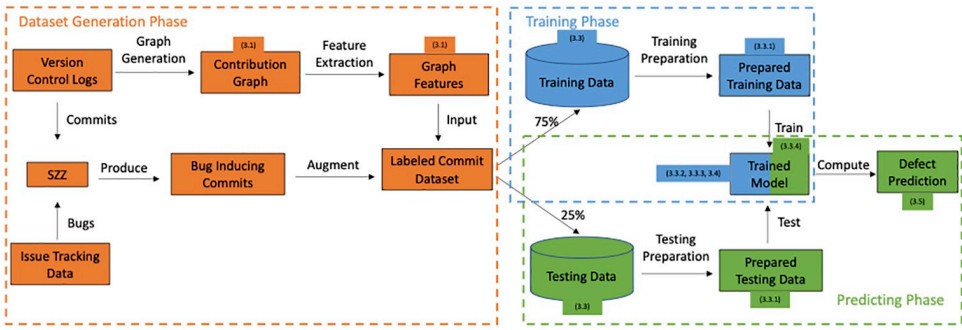

**Fig 1. Graph-based ML JIT defect prediction pipeline.**

the number of changes (i.e., commits) made by developer, $i$, to file, $j$. The graph $\mathcal{G} = (\mathcal{V}, \mathcal{W})$ represents a weighted bipartite graph that captures software changes.

To create the contribution graph, we used changelog metadata entries from `git` logs. Each one of these entries is used to extract the timestamps and information about which developer committed a change to a particular source file. We used Neo4j for storing the bipartite graphs. Neo4j is a high-performance NoSQL database that enables efficient computation of graph-related algorithms, including structural properties and node embeddings, as shown in different applications [51].

**3.1.2 One-mode projection.** We projected the contribution graph into a one-mode projection graph (on the developer side). Specifically, we let $\mathcal{G}_\top = (\mathcal{H}_\top, \mathcal{W}_\top)$ be the top, one-mode projection of $\mathcal{G}$. Two nodes of $\mathcal{G}_\top$ are connected if they have a common neighbor in $\mathcal{G}$, which means the two developers made changes to the same files. In the one-mode projection, we aggregated weights, which results in a weighted, one-mode projection described by the weight matrix, $\mathcal{W}_\top = \{\omega_{uv} : u, v \subseteq \mathcal{H}_\top\}$, where $\omega_{uv} = \sum_{r=1}^{|\mathcal{H}_\perp|} \omega_{ur} + \omega_{vr}$. We extracted graph-based features from the one-mode projection graph using the Neo4j Graph Data Science application programming interface (API). We assumed that the centrality metrics (Setting 1) and community assignments along with node embeddings (Setting 2) in the one-mode projection graph capture the connectivity around both edge endpoints (i.e., developer and file) in the contribution graph. Fig 2 shows an illustration of a toy contribution graph (a) and its one-mode projection (b).

**3.1.3 Edge classification.** Edges in the contribution graph are labeled. Edge labels represent if a particular change is defect-prone or not. Edge classification refers to the task of classifying the edge labels [21]. More formally, consider the set of labeled edges: $\mathcal{W}_\top^\ell \subseteq \mathcal{W}_\top$. Edges $\omega_{ij} \in \mathcal{W}_\top^\ell$ have a binary label, $\ell \in \{0, 1\}$. Edge classification consists on determining the labels of the edges in $\mathcal{W}_\top^u = \mathcal{W}_\top \setminus \mathcal{W}_\top^\ell$.

The key for the edge classification task is to design features for a pair of nodes. We extracted two different types of features from the one-mode projection graphs to train ML models. The first type corresponds to a subset of structural properties. Specifically, we extracted centrality metrics from the nodes. These metrics capture the relative importance of nodes with respect to shared changes in the contribution graph. The second type of feature corresponds to node's community assignments and embeddings. These assignments and embeddings capture more complex, nuanced neighborhood information of nodes to reflect the collaborative structure of software changes. We describe each of them in more detail below.

**3.1.4 Graph structural properties (Setting 1).** We extracted centrality metrics from the nodes in the one-mode projection graph as a proxy for structural properties in the contribution graph. Centrality metrics quantify the relative importance of nodes in the graph [52]. In the context of contribution graphs, centrality metrics identify developers that contribute to many changes in similar modules. We extracted the following centrality metrics:

**3.1.5 degree.** The degree of node $i \in \mathcal{H}_\top$ is the number of edges attached to it. It is quantified as $d(i) = \sum_{(i,j) \in \mathcal{W}_\top} \omega_{ij}$. In the context of contributions graphs, a high degree indicates a developer that has made changes to many modules in conjunction with other developers. In other words, they represent highly collaborative developers that made changes across different modules.

**3.1.6 Betweenness.** The betweenness of node $i \in \mathcal{H}_\top$ is the proportion of geodesic paths that include node $i$. It is quantified as $b(i) = \sum_{i,j,k \in \mathcal{H}_\top} \frac{\sigma_{jk(i)}}{\sigma_{jk}}$, where $\sigma_{jk}(i)$ is the total number of shortest paths that pass through node $i$, and $\sigma_{jk}$ is the total number of shortest paths between

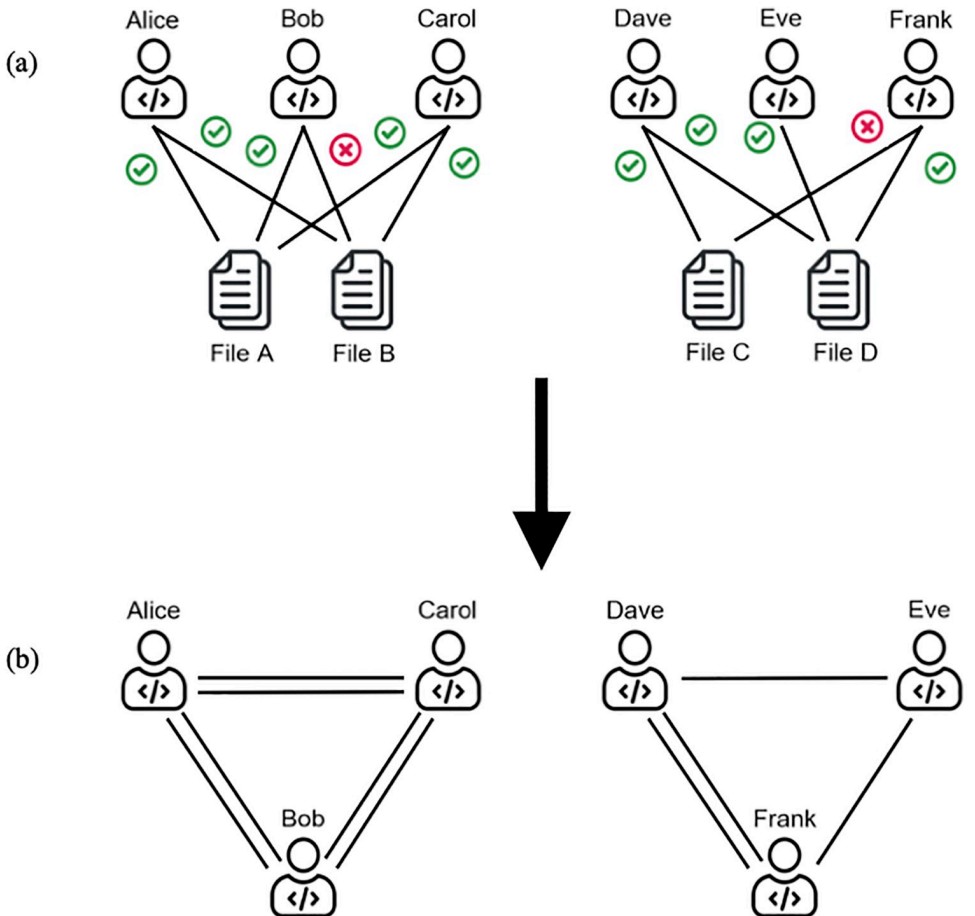

**Fig 2. Contribution graph and its projection.** a) A toy contribution graph. Check marks represent clean changes, whereas cross marks represent defect-prone changes. (b) Corresponding one-mode projection on the developer side. Classification is driven by developer-based features alone as the one-mode projection graph captures the connectivity around both endpoints (i.e., developer and file) in the contribution graph.

nodes $j$ and $k$. Developers with high betweenness are expected to contribute more widely among diverse groups of software modules.

**3.1.7 Closeness.** The closeness of node $i \in \mathcal{H}_\top$ is the average distance from node $i$ to any other nodes in the graph that can be reached from node $i$. It is quantified as $c(i) = \frac{|\mathcal{H}_\top|-1}{\sum_{j\in\mathcal{H}_\top} d(i,j)}$.

Closeness extends the notion of degree to account for distances to any other nodes beyond immediate neighbors. Developers with high closeness are expected to contribute to software modules that are highly dispersed.

**3.1.8 Harmonic.** The harmonic centrality of node $i \in \mathcal{H}_\top$ is a variant of the closeness centrality that can be used to deal with unconnected graphs. It is defined as $h(i) = \frac{(|\mathcal{H}_\top|-1)^{-1}}{\sum_{j\in\mathcal{H}_\top, j\neq i} d(i,j)}$.

**3.1.9 PageRank.** The PageRank of node $i \in \mathcal{H}_\top$ measures each developer's prominence in the contribution graph. PageRank, which was originally conceived to rank web pages based on their importance [53], estimates the stationary probability that a random walker traversing the graph will arrive at a particular node. The stationary probability distribution over all the nodes is quantified by $PR(i) = \frac{1-p}{|\mathcal{H}_\top|} + p\sum_{(i,j)\in\mathcal{W}^\top} \frac{\omega_{ij}PR(j)}{d(j)}$, where $p$ is a damping factor, usually set to

0.85. Because our one-mode developer network does not have directional edges, we treated each directional edge as two directional edges. Developers with high PageRank are the ones that contribute to modules that also received contributions from important developers, who also have a high PageRank.

**3.1.10 Communities and node embeddings (Setting 2).** The community assignment of node $i \in \mathcal{H}_\top$ captures its group identifier. Nodes within the same community are those with a significantly higher number of edges between them as opposed to other nodes in different communities [54]. In particular, let $C = \{c_1, c_2, \ldots, c_{|\mathcal{H}_\top|}\}$ denote a community partition of graph $\mathcal{H}_\top$, thereby indicating the community membership of each node. Meaning, $c_i$ and $c_j$ have the same value if both $i$ and $j$ belong to the same community. Here, we identified communities in the on-mode developer graph using the Louvain algorithm [55]. The Louvain algorithm is based on optimizing the modularity score of each community. The modularity of a community partition quantifies the quality of the nodes' community assignment. The optimization process used in the Louvain algorithm computes how many more densely connected nodes are inside communities compared to a random graph.

We embedded nodes in the graph $\mathcal{G}_\top = (\mathcal{H}_\top, \mathcal{W}_\top)$ in a $d$-dimensional space. This means that every node $i \in \mathcal{G}_\top$ is represented by a unique $d$-dimensional vector that contains the coordinate values of node $i$ in the embedding. In other words, let $f : \mathcal{H}_\top \rightarrow \mathbb{R}^d$ be the mapping function from nodes to a feature representation. Equivalently, $f$ is a matrix of size $|\mathcal{H}_\top| \times d$. The proposed method can be applied using any embedding technique. Here, we used `node2vec` because it has shown robust results for link-related tasks [44]. We adopted commonly used values for the parameters of `node2vec` [20], specifically, a walk length of 80, a context size of 10, in/out of 1.0, return factor of 1.0, and an embedding dimension of 128. Note that these values are already the default in the Neo4j Graph Data Science API.

We summarize the notation used to model the contribution graphs and their extracted features in each Setting in Table 2.

**Table 2. Summary of notation used.**

| Context | Notation | Description |
|---|---|---|
| Contribution graph | $\mathcal{H}_\top$ | The set of top nodes or developers |
| | $\mathcal{H}_\perp$ | The set of bottom nodes or files |
| | $\mathcal{V}$ | The set of contribution graph nodes |
| | $\mathcal{W}$ | The incident matrix of weights |
| | $\mathcal{G} = (\mathcal{V}, \mathcal{W})$ | The weighted bipartite graph capturing software changes |
| | $\mathcal{G}_\top$ | The developer-based one-mode projection |
| | $\mathcal{W}_\top$ | Weighted matrix in the developer-based one-mode projection |
| | $\mathcal{W}_\top^\ell$ | The set of labeled $\mathcal{W}_\top$ edges |
| Setting 1 | $d(i)$ | The degree of node $i$ |
| | $b(i)$ | The betweenness of node $i$ |
| | $c(i)$ | The closeness of node $i$ |
| | $h(i)$ | The harmonic centrality of node $i$ |
| | $PR(i)$ | The PageRank of node $i$ |
| Setting 2 | $C$ | The community partition |
| | $f$ | The node embedding mapping function |

### 3.2 Dataset

We tested the proposed method on the dataset collected by Tian et al. [10]. This dataset contains 18 well-known, open source multi-application projects coded in Java (17) and C++ (1). Note that this dataset tracked nearly the entire development cycle of these projects, as opposed to a limited software release time window [56]. We reported results on 14 of 18 of these repos because four of them (POI, Pig, VELOCITY, and XERCESC) have missing commits. We believe that missing commits may correspond to name changes in the branches of the repo, and we omitted them for that reason. Table 3 summarizes the dataset used. We describe the projects in the dataset in terms of age, kilo (thousands) of lines of code (KLOC), number of changes (# Changes), and early exposed defect ratio used in training (tr.) and testing (te.).

The dataset, which is also available for download [22], focuses on the active middle part of each software effort by trimming off the inactive start and end of each project. The final dataset contains over 85% of the changes spanning only 30% of the original development period. This dataset already contains the annotated, defect-prone changes after running an open-source implementation of the SZZ algorithm, known as *SZZ Unleashed* [29].

When processing this dataset, the focus is on identifying early exposed defects by tracking the time gap between defect-prone changes and defect exposure. A threshold, $\theta$, is used to identify early exposed defects, which are defects that last less than $\theta$. Specifically, $\theta = \min(4\ weeks, 1\% \times (\tau - \tau_0))$, where $\tau_0$ and $\tau$ are the beginning and the end time of the whole project. This helps account for the time span of software projects, which can vary from months to years. We used whether or not a software change contains early exposed defects as the dependent variable for the JIT defect prediction.

Note that the dataset we used here already provides labeled defect introduction fix pairs [10]. However, the proposed framework is flexible enough to be used in other code repos in which the SZZ algorithm can be applied on the required version control logs and labeled defects.

### 3.3 Study setup

We randomly partition the dataset of labeled changes into training sets (75%) and testing sets (25%). The testing data is only used once for computing the performance of the classification

**Table 3. The 14 open-source target projects from the dataset.**

| Project | Description | Age (years) | KLOC | #Changes | Period | Pos. tr. | Neg. tr. | Pos. te. | Neg. te. |
|---|---|---|---|---|---|---|---|---|---|
| ActiveMQ | High-performance messaging server | 13.45 | 38 | 10,213 | 2005–2020 | 43.34% | 56.66% | 43.27% | 56.73% |
| Ant | A Java-based build tool | 8.75 | 339 | 14,387 | 2000–2020 | 21.43% | 78.57% | 21.91% | 78.09% |
| Camel | An open-source integration framework | 11.53 | 75 | 38,563 | 2007–2020 | 17.24% | 82.76% | 17.35% | 82.65% |
| Derby | Java-based relational database engine | 9.42 | 1350 | 8,268 | 2004–2020 | 41.86% | 58.14% | 42.01% | 57.99% |
| Geronimo | An open-source server runtime | 6.79 | 48 | 13,137 | 2003–2020 | 42.66% | 57.34% | 43.15% | 56.85% |
| Hadoop | Open-source distributed computing system | 11.59 | 102 | 16,084 | 2009–2020 | 47.58% | 52.42% | 47.41% | 52.59% |
| HBase | A distributed, scalable, big data store | 5.03 | 413 | 10,509 | 2007–2020 | 0.26% | 99.74% | 0.23% | 99.77% |
| IVY | A project dependencies managing tool | 3.29 | 135 | 2,880 | 2005–2020 | 35.20% | 64.80% | 35.38% | 64.62% |
| JCR | Repo for Java Technology API | 8.82 | 38 | 8,651 | 2004–2020 | 2.87% | 97.13% | 2.84% | 97.16% |
| JMeter | Load test applications/measure performance | 15.89 | 264 | 16,341 | 1998–2020 | 18.04% | 81.96% | 18.09% | 81.91% |
| LOG4J2 | A logging library for Java | 5.37 | 85 | 10,690 | 2010–2020 | 10.96% | 89.04% | 11.35% | 88.65% |
| LUCENE | Full-featured text search engine library | 15.93 | 183 | 31,240 | 2001–2020 | 20.48% | 79.52% | 20.60% | 79.40% |
| Mahout | Linear algebra framework and Scala DSL | 5.52 | 189 | 4,115 | 2008–2020 | 36.57% | 63.43% | 36.76% | 63.24% |
| OpenJPA | Java Persistence API specification | 7.72 | 108 | 4,893 | 2006–2020 | 32.70% | 67.30% | 32.49% | 67.51% |

task. We implemented the proposed framework using the `scikit-learn` [57] and `imbalanced-learn` [58] APIs. After the train/test split, the following steps are performed.

**3.3.1 Data preparation.** We scaled independent variables using min-max normalization. We noticed that the dataset is imbalanced given that the number of defect-prone changes in both training and testing datasets is much smaller than the number of benign changes (see Table 3). We used the Synthetic Minority Oversampling Technique (SMOTE) to handle the imbalance [59]. SMOTE processes each sample in the minority class to generate new synthetic samples along the line by joining them to their $k$-nearest neighbors. We used regular SMOTE with $k = 5$, owing to its simplicity and higher performance. SMOTE can also help increase the framework's ability to classify defective modules [60]. We applied SMOTE only in the training dataset.

**3.3.2 Build model.** We trained prediction models by using the labeled training dataset. The methods used to train the models are described in Section 3.4.

**3.3.3 Select model.** To achieve optimal performance, tuning is performed to find the optimal set of hyperparameters for each model. A grid search is used to consider a small combination of parameters with reasonable values, and a stratified, 10-fold cross validation is implemented to evaluate model performance during this step.

**3.3.4 Apply model.** The optimal defect prediction model is applied on the testing dataset. For each change in the testing dataset, the proposed model predicts whether the change is likely to introduce a defect and then outputs a binary label.

## 3.4 Building prediction models

The graph-based ML models for JIT defect prediction are built using two settings. The first setting leverages features extracted from the centrality properties of the one-mode projection graph (i.e., degree, betweenness, closeness, harmonic, and PageRank). The second setting uses the community assignment and the nodes' embeddings in the one-mode projection graph.

Both settings use three types of classifiers: (1) logistic regression (regression-based classifier), (2) random forest (ML-based classifier), and (3) extreme gradient boosting (XGBoost) (ML-based classifier). These classifiers have been widely used for JIT defect prediction [4, 35, 61]. Each classifier is described below along with their train and test time complexities. Here, $n$ refers to the number of samples, $m$ refers to the number of features, $t$ is the number of trees, $d$ is the height of the trees, and $x$ is the number of non-missing entries in the training dataset [62]. Their default parameters were used unless otherwise noted.

**3.4.1 Logistic regression.** Logistic regression is used for binary classification [63] and models the relationship between one or more independent variables (i.e., extracted from the one-mode projection graph) and a binary dependent variable (i.e., defect-prone or clean changes). The training and testing time complexities of logistic regression are $O(nm)$ and $O(m)$ respectively. We performed a grid search over the inverse of the regularization strength parameter: $C \in [0.01, 0.1, 1.0, 10, 100]$. The optimal value is 100. The training and testing time complexities of logistic regression are $O(nm)$ and $O(m)$, respectively.

**3.4.2 Random forest.** Random forest is an ensemble method that leverages a large number of decision trees [64]. Each of these trees focuses on a random subset of features. When reporting a decision, trees may report different results. The random forest then aggregates each of the results from the trees to make a final decision. The training and testing time complexities of random forest are $O(tn\log nm)$ and $O(mt)$ respectively. We performed a grid search of trees in the forest parameter: `n_estimators` $\in [10, 100, 1000]$. We found that the optimal value is 100.

**3.4.3 XGBoost.** XGBoost is an implementation of gradient-boosted decision trees that is designed for speed and performance [65]. Boosting is an ensemble method in which new models are added iteratively to improve performance. The process stops at diminishing returns. Gradient boosting is used to create new models that predict the error of previous models; they are then added together to make a final decision. Gradient boosting uses gradient descent to minimize errors when adding new models. The training and testing time complexities of XGBoost are $O(tdx\log n)$ and $O(td)$ respectively. We performed a grid search of the learning rate parameter, `learning_rate` $\in$ [0.001, 0.01, 0.1], and of the number of trees in the forest parameter, `n_estimators` $\in$ [10, 100, 1000]. We found that the optimal set of values is 0.01 for the learning rate and 1000 for the number of trees.

## 3.5 Evaluation metrics

We used confusion matrix–based metrics to compute the performance of the proposed methods, which have been widely employed for JIT defect prediction [10, 66]. The basis for comparison is counting the number of code changes that were labeled as true positives (TP), false positives (FP), false negatives (FN), and true negatives (TN). We focus on *Precision*, defined as $\frac{TP}{TP+FP}$, which gives the likelihood that a detected change is defect-prone; *Recall*, defined as $\frac{TP}{TP+FN}$, which gives the likelihood that a defect-prone change is detected; *F1 score*, defined as $2 \times \frac{precision \times recall}{precision+recall}$, which combines precision and recall to provide a balanced view between them; and the Matthews correlation coefficient (MCC), defined as $\frac{TP \times TN - FP \times FN}{\sqrt{(TP+FP) \times (TP+FN) \times (TN+FP) \times (TN+FN)}}$, which is the Pearson correlation for a contingency table [67]. MCC takes values in the range [−1, 1], with extreme values −1 and + 1 reached in the case of perfect misclassification and perfect classification, respectively. MCC equals 0 is equivalent to the expected value for the coin-tossing classifier. We include the MCC metric because the above metrics overemphasize the positive class while not putting much emphasis on the negative class, which is also important for defect prediction. Note, however, that among the 14 repos that we analyzed, they have different levels of imbalance with only one of them (HBase in Table 2) showing extreme imbalance (i.e., less than 1% for the minority class [68]). Therefore, we report results by using precision, recall, and F1 scores in addition to MCC. We let the reader explore the literature [69, 70] for a more thorough discussion of the advantages of MCC over other classification metrics. We evaluate the performance of our proposed framework against a state-of-the-art baseline that uses software-level characteristics as features of a random forest classifier [10].

To compare the performance of the proposed framework using different classifiers over a range of detection thresholds, we used the Precision-Recall (PR) curve. We chose the PR curve instead of the commonly used receiver operating characteristic curve because the PR curve is better suited for handling highly imbalanced datasets [71, 72]. We reported the results of precision, recall, and F1 score based on the optimal threshold obtained from the PR curve for the F1 score. We report results for the MCC based on the default threshold of 0.5. The PR curve also allows for a head-to-head method comparison (independent of thresholds) based on the area under the PR curve (AUC-PR). Higher percentages indicate better overall performance. Note that this metric was not computed in the state-of-the-art baseline. Given that our results are aggregated over different datasets, for comparison we report the mean value and the average rank of each classifier on each metric across the 14 datasets (shown in the *Avg. R* column in Table 4).

## 4 Results

This section details the performance of the proposed framework based on graph-based ML for JIT defect prediction.

**Table 4. Classification results for both settings with the two best performing classifiers shown in bold text.**

| | | Precision | | Recall | | F1 score | | MCC | | AUC-PR | |
|---|---|---|---|---|---|---|---|---|---|---|---|
| | | Mean | Avg. R | Mean | Avg. R | Mean | Avg. R | Mean | Avg. R | Mean | Avg. R |
| Setting 1 | Logistic Regression | 0.6050 | 2.57 | 0.6005 | 2.64 | 0.5871 | 2.86 | 0.3393 | 2.64 | **0.8074** | 2.29 |
| | Random Forest | **0.7359** | **2.00** | **0.8241** | **1.86** | **0.7724** | **1.79** | **0.5316** | **1.86** | 0.8022 | **2.07** |
| | XGBoost | **0.7341** | **1.43** | **0.8239** | **1.50** | **0.7755** | **1.36** | **0.5234** | **1.50** | 0.7993 | **1.64** |
| Setting 2 | Logistic Regression | 0.6407 | 2.50 | 0.7773 | 2.36 | 0.6950 | 2.36 | 0.3231 | 2.64 | **0.8151** | 1.93 |
| | Random Forest | **0.7343** | **2.07** | **0.8237** | **2.00** | **0.7714** | **2.00** | **0.5215** | **2.07** | 0.8015 | 2.21 |
| | XGBoost | **0.7418** | **1.43** | **0.8235** | **1.64** | **0.7748** | **1.64** | **0.5234** | **1.29** | 0.8061 | 1.86 |
| Baseline [10] | Random Forest | 0.4673 | 1.03 | 0.7644 | 1.03 | 0.3083 | 1.03 | 0.5141 | 1.03 | — | — |

## 4.1 Comparison of graph-based ML with the baseline

We applied the two settings of graph-based ML classifiers and measured the performance in the classification task by using the mean and average ranks of Precision, Recall, F1 score, AUC-PR, and MCC across the 14 repositories of code. These metrics are defined in Section 3.5. Table 4 summarizes the results based on the performance evaluation in the two settings (see Section 3.4 for details). We observe that the graph-based ML classifiers based on random forest and XGBoost (in both settings) outperform the baseline that uses a random forest classifier in terms of F1 score and MCC. The baseline reports lower precision on average than the proposed framework in both settings. This means that the proposed framework produces fewer FPs, even when using a simpler classifier, such as logistic regression. Likewise, the baseline reports lower average recall results, suggesting that the proposed framework detects more defect-prone changes (TPs) than the baseline, on average. The F1 scores obtained by the proposed framework show a relative improvement of at least 90% (from 30.83% to 58.71%) for the logistic regression classifier in Setting 1 and as much as 152% (from 30.83% to 77.55%) for the XGBoost classifier in Setting 1. Finally, the MCC scores obtained with the proposed framework are relatively higher than the baseline for random forest and XGBoost classifiers (i.e., 3% [from 51.41% to 53.16%] and 2% [from 51.41% to 52.34%] in Setting 1 and 1% [from 51.41% to 52.15%] and 2% [from 51.41% to 52.34%] in Setting 2). These MCC scores reflect a moderate positive relationship (between 0.3 and 0.7 based on [73]). Yet, this does not hold for the logistic regression classifier. Note, however, that the F1 score and MCC results are concordant when using random forest and XGBoost classifiers, meaning that we obtain consistent preferred classifiers regardless of whether we use an F1 score or MCC score when comparing with the baseline [70]. This provides empirical evidence that the proposed framework performs better than the state-of-the-art baseline when using classifiers of similar complexity.

## 4.2 Comparison of graph-based ML classifiers

We also compare the performance of different classifiers in each setting by using the mean of AUC-PR. Under Setting 1, the three classifiers perform similarly with a slight advantage of logistic regression (80.74%) over random forest (80.22%) and XGBoost (79.93%). The relative performance increase of logistic regression based on AUC-PR is at most 1% over XGBoost. Note, however, that the XGBoost classifier tends to perform best in terms of average ranking across the 14 repos (i.e., [1.64] over random forest [2.07] and logistic regression [2.29]). Under Setting 2, the best performing classifier based on average AUC-PR is logistic regression (81.51%) followed by XGBoost (80.61%) with a relative performance increase of at most 1%. However, like in Setting 1, the best performing classifier based on rankings is again XGBoost

(1.86) over logistic regression (1.93) and random forest (2.21). This suggests that across repos, XGBoost consistently has the best performance despite outliers.

In general, we observe that the performance of graph-based ML classifiers is similar in each setting based on mean AUC-PR. The observed differences based on rankings are of at most 0.65 (from 1.64 of XGBoost to 2.29 of logistic regression) in Setting 1 and 0.35 (from 1.86 of XGBoost to 2.21 of random forest) in Setting 2. This suggests that the exclusive use of structural features from the contribution graphs in the classification task (i.e., Setting 1) benefits from a more complex classification function derived from more advanced classifiers, such as random forest and XGBoost. In contrast, under Scenario 2, the more complex features make the classifier task less determinant. This observation is also corroborated by the reported average rankings. Recall that in Setting 2, we use community assignments and node embeddings of length 128.

## 5 Discussion

JIT defect prediction is at the core of software quality assurance efforts. Here, we proposed using graph-based ML to improve JIT defect prediction. To do so, we constructed contribution graphs (or bipartite graphs made of developers and source files) and framed the JIT defect prediction challenge as an edge classification problem, in which the objective was to classify defect-prone edges in the contribution graph. We extracted features from a projected version of the contribution graph by computing centrality measures of the nodes (Setting 1) and community assignment and node embeddings (Setting 2). We showed that the relative performance increase in the JIT defect prediction task is 152% for F1 score and 3% for MCC over the state-of-the-art baseline when using Setting 1.

We validated the effectiveness of the proposed approach by performing JIT defect prediction on 14 open-source software projects. We assessed the predictive power of graph-based features on edge classification using logistic regression, random forest, and XGBoost classifiers. Overall, we found that using graph-based features improved classification accuracy over traditional repo-based features, such as those related to size, purpose, and history of code bases. Comparing Setting 1 and Setting 2, we find that they tend to produce similar results encoded in almost negligible differences in the mean of classification results. Comparing classification models based on rankings, XGBoost outperformed random forest and logistic regression for the code bases we tested in our approach—at the expense of computational complexity. We conclude, however, that by using the features derived from Setting 2, the decision of what classifier to use for better performance is less important and produces negligible differences. We are sharing the data [22] and code [23] used in this research so that our results can be reproduced.

The proposed graph-based ML framework is effective at improving the detection performance of JIT defect prediction. Having noted the potential, we are also aware of some limitations of the proposed work.

### 5.1 Accuracy of the SZZ algorithm

Our analysis is based on the assumption that code changes are correctly labeled by the SZZ algorithm. We acknowledge that SZZ can mislabel changes, thereby impacting the results of JIT defect prediction [74].

### 5.2 Extracting features from static graphs

We study a snapshot of a contribution graph, but this is inherently dynamic. This implies that the structural features and the node embeddings extracted using `node2vec` need to be

recomputed when the graph changes so that they do not generalize to unseen nodes and/or edges. Thus, we did not assess, until what extent, the temporal information of developers changing files may be useful for the prediction of software defects.

### 5.3 Strict focus in network topology

Our analysis relies on the structure of the contribution graphs to generate classification features. We do not consider node and/or edge attributes, such as experience, programming proficiency, and education, in the case of nodes; and we do not consider characteristics of code changes, such as size, diversification of changes, and time, in the case of edges.

### 5.4 Feature-importance assessment

We report results based on average performance and rankings for precision, recall, F1 score, and AUC-PRC. This compact performance representation can hinder specific performance details for particular code bases and the effect of model features (for both Setting 1 and Setting 2).

### 5.5 Classifier's complexity and interpretability

Performance gains obtained by using the graph-based ML classifiers are fueled by the rich set of features extracted from the contribution graphs. However, classification results are also dependent on the type of classifier used. In that respect, both complexity and interpretability are advantages of the simpler models, such as logistic regression, as opposed to more robust models, such as random forest and XGBoost, despite superior performance.

### 5.6 Use of default graph algorithms' parameter values

We run the graph algorithms to extract features from the one-mode projection graphs by using their default parameters. See related documentation of graph algorithm parameters in [75]. Thus, we do not optimize results by tuning these parameters.

### 5.7 Use of AUC-style metrics to compare classifiers

Note that AUC-style metrics are based on a family of possible classifiers at different thresholds instead of a specific classifier. Because in practice only a single classifier can be deployed, we also include other metrics that focus on a single threshold (i.e., F1 score and MCC) to better qualify the best performing classifier.

## 6 Conclusion

In this paper, we show the potential of graph-based ML for JIT defect prediction. The proposed framework outperforms the state-of-the-art in JIT defect prediction with a higher average F1 score (152%) and higher average MCC (3%) across 14 open-source projects. Our contribution focus on characterizing the process of building software using a contribution graph (a bipartite graph of developers and source files) by capturing the nuanced structure of code collaborations. In the contribution graph, edges represent code changes made by developers. Our proposed framework leverages centrality metrics (Setting 1) and node's community assignments and embeddings (Setting 2) extracted from the contribution graph for edge classification. Relying on this abstraction, we detailed a classification framework that can decide whether a change is defect-prone or not.

Future work will include examining the effectiveness of the proposed approach using inductive embedding frameworks, such as GraphSAGE [76]. The approach proposed in this

work would benefit from an inductive framework that can efficiently generate node embeddings from previously unseen graph data by aggregating features from node-local neighborhoods, including node attributes. In addition, future work could include benchmarking the effectiveness of the proposed framework with more diverse codebases to further improve the generalizability of the proposed method. A working prototype for JIT defect prediction using the principles described in this paper seems feasible because the tools are available and ready to use in the Neo4j Graph Data Science API.

## Acknowledgments

We are grateful to the reviewers and the editor for their constructive input that help us to improve our manuscript. Pablo Moriano thanks David Womble and Sudip Seal for their guidance.

## Author Contributions

**Conceptualization:** Pablo Moriano.

**Data curation:** Jonathan Bryan.

**Formal analysis:** Pablo Moriano.

**Funding acquisition:** Pablo Moriano.

**Investigation:** Jonathan Bryan, Pablo Moriano.

**Methodology:** Pablo Moriano.

**Project administration:** Pablo Moriano.

**Resources:** Pablo Moriano.

**Software:** Jonathan Bryan.

**Supervision:** Pablo Moriano.

**Validation:** Pablo Moriano.

**Visualization:** Jonathan Bryan, Pablo Moriano.

**Writing – original draft:** Pablo Moriano.

**Writing – review & editing:** Pablo Moriano.

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
