## [Decision Letter · Decision Letter 0]

14 Dec 2022

PONE-D-22-18053Graph-based machine learning improves just-in-time defect predictionPLOS ONE

Dear Dr. Moriano,

Thank you for submitting your manuscript to PLOS ONE. After careful consideration, we feel that it has merit but does not fully meet PLOS ONE’s publication criteria as it currently stands. Therefore, we invite you to submit a revised version of the manuscript that addresses the points raised during the review process.

We look forward to receiving your revised manuscript.

Kind regards,

Orawit Thinnukool, Ph.D.

Academic Editor

PLOS ONE

Journal Requirements:

“This research was sponsored in part by Oak Ridge National Laboratory's (ORNL's) Laboratory Directed Research and Development program. Pablo Moriano acknowledges support from ORNL's Artificial Intelligence initiative. The funders had no role in study design, data collection and analysis, decision to publish, or preparation of the manuscript.”

“This research was sponsored in part by Oak Ridge National Laboratory’s (ORNL’s) 503 Laboratory Directed Research and Development program. Pablo Moriano acknowledges 504 support from ORNL’s Artificial Intelligence initiative.”

“This research was sponsored in part by Oak Ridge National Laboratory's (ORNL's) Laboratory Directed Research and Development program. Pablo Moriano acknowledges support from ORNL's Artificial Intelligence initiative. The funders had no role in study design, data collection and analysis, decision to publish, or preparation of the manuscript.”

Additional Editor Comments (if provided):

Dear Author

The manuscript Graph-based machine learning improves just-in-time defect prediction by investigating very important and practice problems. The author presented a sufficient introduction and research motivation as well as contribution. The literature review is sufficient. The proposed methods are new and innovative.

In my point of view, the idea of this paper is interesting and has some novelty. The paper is clearly presented and easy to follow assuming the reader has some background knowledge.

Although I found that your paper has merit, it is not acceptable to publish in its present form. Please revise the manuscript according to the reviewers' comments and upload the revised file within 30 days.

Please find the reviewer's comments at the end of this message;

The evaluation is also good. However, the paper can be improved in the following way.

Major:

1. Though the authors have updated the manuscript, a few comments are not still addressed. please consider the following comments and revise the paper :

1. Abstract: Read the complete abstract and try to write direct, simple, and straight sentences e.g., “The main objective ……

2.Open science? Although a link is provided to the data (provided by another research group/Ning Li) there is no code or details for instance on how the holdout samples were undertaken. This means your analysis could not be reproduced. Please provide your python code etc.

3.There does seem a slight tendency to cherry pick results to "big up" their method. This isn't helpful or necessary. It would be better to either give the range of possible results and/or typical improvement.

4. The reviwers concern about the classification performance evaluation. The authors note the problems relating to imbalanced datasets and for that reason advocate AUC-PRC (fair enough) but then extensively quote F1 performance stats which seems odd (e.g., l438).

5. Another problem with AUC style stats is they are based on entire families of classifier (many of which are uninteresting in a practical sense) so unless classifier x strictly dominates y knowing AUC_x > AUC_y is not informative. There are also other concerns (see [1-2])

6. I'd consider using Matthews correlation coefficient/phi or Youden's J (which compares against a guessing strategy).

7.Lack of tuning may be misleading (l480-2). I appreciate you've not tuned any method but it seems a bit unrealistic.

8. The paper should have more case studies related to work.

9. Literature work should have a table in which each paper's method, problem, constraints, and research should be analyzed in detail in tabular form.

10.The research findings and limitations must be defined before the conclusion of the work.

11.Authors mentioned "The core of our contribution is problem 65

formulation. In particular, we leverage contribution graphs to extract graph-related 66

features that inform classification models when classifying defect-prone changes."I do not understand if the contribution is more on data classification modeling or proposing new feature vector OR new framework

Please Elaborate.

12.Expand the comparison with state of art work (at least 3-4 works) not only work [10].

Minor:

1. English should be extensively revised and corrected.

2.Abstract (and l486): "by as much as 46.72%" but this isn't the representative case. I think the authors can reframe this claim in a more reasonable way without losing the impact or value of their work. We're not in the advertising business ;-)

l398: lower -> fewer (being a bit pedantic here!)

REFERENCES:

[1] Hand, D. (2009). Measuring classifier performance: a coherent alternative to the area under the ROC curve. Machine Learning, 77, 103--123. https://doi.org/10.1007/s10994-009-5119-5

[2] Powers, D. (2012). The Problem of Area Under the Curve International Conference on Information Science and Technology (ICIST), Wuhan.

3.The notations should have a particular table.

4. The time complexity of each method must be defined in the paper.

5. -Reorganize the methodology section into phases with one main diagram that represent each phase as well as each step into distinct phase.

6.-Figure 1 and 2 are missing.

Reviewers' comments:

Reviewer's Responses to Questions

**Comments to the Author**

1. Is the manuscript technically sound, and do the data support the conclusions?

Reviewer #1: Partly

Reviewer #2: Yes

Reviewer #3: Partly

2. Has the statistical analysis been performed appropriately and rigorously? 

Reviewer #1: N/A

Reviewer #2: Yes

Reviewer #3: Yes

3. Have the authors made all data underlying the findings in their manuscript fully available?

Reviewer #1: Yes

Reviewer #2: Yes

Reviewer #3: Yes

4. Is the manuscript presented in an intelligible fashion and written in standard English?

Reviewer #1: Yes

Reviewer #2: Yes

Reviewer #3: Yes

5. Review Comments to the Author

Reviewer #1: STRENGTHS:

The idea is interesting and has some novelty.

The paper is clearly presented and easy to follow assuming the reader has some background knowledge.

WEAKNESSES:

Open science? Although a link is provided to the data (provided by another research group/Ning Li) there is no code or details for instance on how the holdout samples were undertaken. This means your analysis could not be reproduced. Please provide your python code etc.

There does seem a slight tendency to cherry pick results to "big up" their method. This isn't helpful or necessary. It would be better to either give the range of possible results and/or typical improvement.

OTHER COMMENTS:

I'm concerned about the classification performance evaluation. The authors note the problems relating to imbalanced datasets and for that reason advocate AUC-PRC (fair enough) but then extensively quote F1 performance stats which seems odd (e.g., l438).

Another problem with AUC style stats is they are based on entire families of classifier (many of which are uninteresting in a practical sense) so unless classifier x strictly dominates y knowing AUC_x > AUC_y is not informative. There are also other concerns (see [1-2])

I'd consider using Matthews correlation coefficient/phi or Youden's J (which compares against a guessing strategy).

Lack of tuning may be misleading (l480-2). I appreciate you've not tuned any method but it seems a bit unrealistic.

MINOR:

Abstract (and l486): "by as much as 46.72%" but this isn't the representative case. I think the authors can reframe this claim in a more reasonable way without losing the impact or value of their work. We're not in the advertising business ;-)

l398: lower -> fewer (being a bit pedantic here!)

REFERENCES:

[1] Hand, D. (2009). Measuring classifier performance: a coherent alternative to the area under the ROC curve. Machine Learning, 77, 103--123. https://doi.org/10.1007/s10994-009-5119-5

[2] Powers, D. (2012). The Problem of Area Under the Curve International Conference on Information Science and Technology (ICIST), Wuhan.

Reviewer #2: The manuscript Graph-based machine learning improves just-in-time defect prediction by investigating very important and practice problems. The author presented a sufficient introduction and research motivation as well as contribution. The literature review is sufficient. The proposed methods are new and innovative. The evaluation is also good. However, the paper can be improved in the following way.

1. The paper should have more case studies related to work.

2. The notations should have a particular table.

3. The time complexity of each method must be defined in the paper.

4. Literature work should have a table in which each paper's method, problem, constraints, and research should be analyzed in detail in tabular form.

5.The research findings and limitations must be defined before the conclusion of the work.

Reviewer #3: Many drawbacks are existed into presented paper such as:

-Authors mentioned "The core of our contribution is problem 65

formulation. In particular, we leverage contribution graphs to extract graph-related 66

features that inform classification models when classifying defect-prone changes."I do not understand if the contribution is more on data classification modeling or proposing new feature vector OR new framework. Elaborate.

-Reorganize the methodology section into phases with one main diagram that represent each phase as well as each step into distinct phase.

-Figure 1 and 2 are missing.

-Expand the comparison with state of art work (at least 3-4 works) not only work [10].

6. PLOS authors have the option to publish the peer review history of their article (what does this mean?). If published, this will include your full peer review and any attached files.

Reviewer #1: **Yes: **Martin Shepperd

Reviewer #2: **Yes: **Abdullah Lakhan

Reviewer #3: No

---

## [Author Response · Author response to Decision Letter 0]

16 Feb 2023

The handling editor of this submission is Orawit Thinnukool. Response to the reviewers was attached as a separate PDF file.

---

## [Editor Report · Decision Letter 1]

23 Mar 2023

Graph-based machine learning improves just-in-time defect prediction

PONE-D-22-18053R1

Dear Dr. Moriano,

We’re pleased to inform you that your manuscript has been judged scientifically suitable for publication and will be formally accepted for publication once it meets all outstanding technical requirements.

Kind regards,

Orawit Thinnukool, Ph.D.

Academic Editor

PLOS ONE

Additional Editor Comments (optional):

I am confident that the paper is now ready for publication.
---

## [Editor Report · Acceptance letter]

3 Apr 2023

PONE-D-22-18053R1 

Graph-based machine learning improves just-in-time defect prediction 

Dear Dr. Moriano:

I'm pleased to inform you that your manuscript has been deemed suitable for publication in PLOS ONE. Congratulations! Your manuscript is now with our production department. 

Kind regards, 

on behalf of

Assistant Professor Orawit Thinnukool 

Academic Editor

PLOS ONE